# DNA Methylation and Histone Modification in Low-Grade Gliomas: Current Understanding and Potential Clinical Targets

**DOI:** 10.3390/cancers15041342

**Published:** 2023-02-20

**Authors:** Ahmad Ozair, Vivek Bhat, Reid S. Alisch, Atulya A. Khosla, Rupesh R. Kotecha, Yazmin Odia, Michael W. McDermott, Manmeet S. Ahluwalia

**Affiliations:** 1Miami Cancer Institute, Baptist Health South Florida, Miami, FL 33176, USA; 2Faculty of Medicine, King George’s Medical University, Lucknow 226003, India; 3St. John’s Medical College, Bangalore 560034, India; 4Department of Neurosurgery, University of Wisconsin-Madison, Madison, WI 53792, USA; 5Herbert Wertheim College of Medicine, Florida International University, Miami, FL 33199, USA; 6Miami Neuroscience Institute, Baptist Health South Florida, Miami, FL 33176, USA

**Keywords:** methylation, methylomics, G-CIMP, MGMT, DNMT, ATRX, H3K27M, CpG island, tumor suppressor, methyltransferases, histone acetylation

## Abstract

**Simple Summary:**

Brain tumors comprise a large, varied group, with gliomas being the most common malignant tumors arising in the brain. This state-of-the-art review discusses the role of epigenetics in low-grade gliomas, i.e., those gliomas which are typically less invasive and have better survival rates than their high-grade counterparts. This paper is a summary of the current paradigms in DNA methylation and histone modification in low-grade gliomas, with their integration into the recently published WHO Classification for CNS Tumors, Fifth Edition. This paper, targeted towards a clinical audience, also describes the role of DNA methylation and histone modification in pathogenesis, clinical behavior, and outcomes of low-grade gliomas, with an emphasis on the potential therapeutic targets in associated cellular biomolecules, structures, and processes.

**Abstract:**

Gliomas, the most common type of malignant primary brain tumor, were conventionally classified through WHO Grades I–IV (now 1–4), with low-grade gliomas being entities belonging to Grades 1 or 2. While the focus of the WHO Classification for Central Nervous System (CNS) tumors had historically been on histopathological attributes, the recently released fifth edition of the classification (WHO CNS5) characterizes brain tumors, including gliomas, using an integration of histological and molecular features, including their epigenetic changes such as histone methylation, DNA methylation, and histone acetylation, which are increasingly being used for the classification of low-grade gliomas. This review describes the current understanding of the role of DNA methylation, demethylation, and histone modification in pathogenesis, clinical behavior, and outcomes of brain tumors, in particular of low-grade gliomas. The review also highlights potential diagnostic and/or therapeutic targets in associated cellular biomolecules, structures, and processes. Targeting of MGMT promoter methylation, TET-hTDG-BER pathway, association of G-CIMP with key gene mutations, PARP inhibition, IDH and 2-HG-associated processes, TERT mutation and ARL9-associated pathways, DNA Methyltransferase (DNMT) inhibition, Histone Deacetylase (HDAC) inhibition, BET inhibition, CpG site DNA methylation signatures, along with others, present exciting avenues for translational research. This review also summarizes the current clinical trial landscape associated with the therapeutic utility of epigenetics in low-grade gliomas. Much of the evidence currently remains restricted to preclinical studies, warranting further investigation to demonstrate true clinical utility.

## 1. Introduction

Gliomas are a heterogenous group of central nervous system (CNS) tumors that are grouped based on their common origin from glial or precursor cells [1,2]. Gliomas include entities such as glioblastoma, astrocytoma, oligodendroglioma, ependymoma, and mixed gliomas amongst others. Taken together, they comprise over 60% of all primary brain tumors and nearly 25% of all malignant brain neoplasms [1,3,4,5]. 

Gliomas have been conventionally classified through Grades I-IV (now using 1–4) using the World Health Organization (WHO) schema, with low-grade gliomas typically referring to tumors belonging to Grade 1 or 2, even though some authors have infrequently referred to Grade 3 tumors as LGGs [1,3,5,6,7]. 

To discuss DNA methylation in LGGs, it is essential to (A) first recognize which entities are classified as LGGs currently, as their neuropathological classification has evolved in the last two decades, and (B) have a broad understanding of methylation processes. In general, Grade I gliomas, such as pilocytic astrocytoma, are typically localized, have low invasion potential, and remain amenable to surgical resection [1,8]. Grade 2 gliomas, also called diffuse LGGs (DLGGs), are more locally invasive and require adjuvant strategies for their curative therapy [1,2,4,5,6,8,9]. While the focus of the classification of gliomas has historically been on clinicopathological attributes, the recently released fifth edition of the CNS tumor classification (WHO CNS5) now characterizes brain tumors, including gliomas, using an integration of histological and molecular features, including DNA methylation [5]. 

## 2. Current Status of LGGs in the WHO Classification

Historically, gliomas were classified primarily based on their histologic attributes [1,3]. This practice continued until the 2007 WHO classification, which recognized seven different types of gliomas, based on differentiation along astrocytic and/or oligodendroglial lineages [10]. Further prognostic entities were later defined based on the histologic grading, with cellular features of mitoses and necroses associated with both higher grades and worse prognosis [10]. However, this classification system suffered from significant intra-observer and inter-observer variability, along with a lack of clarity regarding reproducible methods. 

With advances in molecular analysis, glioma classification has undergone a paradigm shift, with significant molecular heterogeneity reported among each histologic type of glioma [1,6,11,12]. One such seminal advance was the discovery of mutations in the isocitrate dehydrogenase (IDH) 1 and 2 genes, with IDH1/2 mutations identified in over 70% of LGGs [13]. Furthermore, IDH1/2-mutant (IDHmt) tumors were found to have a demonstrably better prognosis than IDH1/2-wild type (IDHwt). In 2015, a study utilizing The Cancer Genome Atlas (TCGA) analyzed 293 LGGs and identified an additional molecular marker—the loss of chromosomes 1p and 19q—allowing subclassification into three prognostically distinct groups. Arranged from best to worst prognosis, LGGS can be fundamentally ordered into (A) IDH-mutant (IDHmt) LGGs with 1p/19q chromosomal codeletion, e.g., oligodendrogliomas, which are associated with gene mutations of Telomerase Reverse Transcriptase (TERT); (B) IDHmt LGGs without 1p/19q chromosomal codeletion, e.g., astrocytomas that are typically associated with mutations in Tumor Protein 53 (TP53) and ATP-Dependent Helicase ATRX (ATRX); and (C) IDH wild-type (IDHwt) LGGs [14]. Subsequent studies elucidated genetic signatures unique to each of these three groups [15,16].

Recognizing these advances, the WHO 2016 classification of gliomas emerged, which utilized a combination of histologic and molecular signatures for classification [17]. Here, six separate entities of glioma were identified, each with a unique molecular signature. While this was a welcome step, one persistent limitation was the continued reliance on ‘brisk’ mitotic activity to distinguish Grade 3 from Grade 2 gliomas, requiring subjective counting of specimens, something that was compounded by the fact that mitotic activity had little significance in IDHmt LGGs [18]. 

The most recent, fifth edition of the WHO Classification of Tumors of the Central Nervous System (WHO CNS5) took this one step further by incorporating the recommendations from the Consortium to Inform Molecular and Practical Approaches to CNS Tumor Taxonomy (cIMPACT-NOW) [14,19,20,21,22], along with the landmark DNA methylation-based classification of CNS tumors published in Nature [12]. The WHO CNS5 uses an integrated histo-molecular assessment, prioritizing genetic and molecular alterations, which were emphasized for several tumor types [5]. A summary of the view of the WHO CNS5 has been provided in Figure 1.

WHO CNS5 utilizes a hybrid approach with regard to tumor grouping [23]. While some tumor groups still find a lack of utilization for any molecular testing requirements such as meningiomas, several new types and subtypes are primarily characterized by molecular features such as medulloblastoma and ependymomas [5]. Gliomas currently fall under the group of “Gliomas, Glioneuronal and Neuronal Tumors”. The grading of gliomas, now done using WHO Grade 1–4 instead of Grade I–IV, is to be based on a combination of histologic and molecular features [5]. Gliomas have also been separated into pediatric-type and adult-type, thus reorganizing and grouping entities with common genetic alterations (Table 1). Gliomas were also rearranged accounting for their prevalent genetic mutations, especially IDH 1/2 mutation (better prognosis), 1p/19q codeletion (better prognosis), and CDKN2A/B homozygous deletion (worse prognosis). Grading is now to be done within individual tumor types, instead of across tumor types. Perhaps the most landmark change for clinicians was the change in classification of glioblastomas (GBMs). As per WHO CNS5, GBM includes only IDH-wild type entities, while previously GBMs included both IDHmt (10%) and IDHwt (90%) [24].

As per WHO CNS5, diffuse astrocytic tumors can now be classified as Grade 2 (i.e., LGG), Grade 3, or Grade 4, the latter two being high-grade gliomas (HGGs). Diffuse astrocytic tumors with IDHwt, i.e., baseline more aggressive than IDHmt, that lack GBM-specific histology but have at least one of three particular genetic alterations would also be classified as GBMs [5]. These specific alterations are: (1) TERT promoter mutations (TERT-pmt), (2) EGFR gene amplifications, and/or (3) loss of chromosome 10 (+7/−10) [5,23]. On the other hand, IDHmt astrocytomas with CDKN2A/B homozygous deletions and related alterations can now be classified as WHO Grade 4, even if histologically lacking necroses or microvascular proliferation [5]. Thus, IDH mutation testing has become a key requirement for appropriate classification into LGG or HGG [23]. The characterization of methylomic attributes was added to diagnostic criteria, albeit as “desirable characteristics”, acknowledging the general inaccessibility of these tools [25]. 

While recognizing newer or updated entities in the new classification, it is also essential to note that low-grade gliomas (LGG), in particular astrocytomas, can transform into higher-grade tumors or display more aggressive behavior after some time [2]. Nearly 70% of diffuse LGGs transform into a higher-grade type [26,27]. This is likely the result of the gradual accumulation of genetic and epigenetic alterations, which together allow cellular replication to take place in an unrestrained fashion. Epigenetic alterations in cancer cells have been demonstrated to increase genomic fragility, increase angiogenic capabilities, decrease the attribute of cellular adhesion, permit entry into the cell cycle, help avoid apoptosis and lead to defects in DNA repair, as further examined below [28].

## 3. Overview of DNA Methylation and Demethylation

The importance of epigenetic processes in the clinical neurosciences may be amply demonstrated in the role of DNA methylation patterns in the physiological regulation of differentiation, in particular, through cellular, spatial and temporal specificities [29,30]. Notably, epigenetic deregulation has also been included amongst the updated hallmarks of cancer [31,32]. In the cancer cell, it acts in both a standalone fashion and synergistically with other genetic changes, in driving neoplastic evolution [29,30,31,32]. However, despite substantial advances in the understanding and the utility of investigating methylomics of various malignancies, considerably less headway has been made in the clinical utilization of epigenetics in brain tumors, especially in less aggressive tumors such as low-grade gliomas [30]. DNA methylation has been the most widely studied and most clinically explored epigenetic change in gliomas [28]. Given the complexity of the cellular processes involved, a brief review for clinicians of processes involved in DNA methylation follows. 

The cellular DNA, including that of the cancer cell, is constructed out of four elements (DNA bases), namely, adenine (A), thymine (T), guanine (G), and cytosine (C). While adenine and guanine are purines, thymine and cytosine are pyrimidines. Base pairing occurs between AT and GC, but a methylated cytosine, with its corresponding CG base pairing, may undergo deamination to form a thymine. 

DNA methylation is a process by which methyl (CH3-) groups are added to the DNA bases, to allow for an additional layer of regulation of gene expression. This modification can change the activity of a DNA segment without changing the underlying sequence. DNA methylation typically occurs on cytosine bases, leading to the formation of 5-methylcytosine, often called the ‘fifth DNA base’. It is estimated that 3–6% of cytosine bases in human cells carry methyl groups [28], where it is especially predominant in repetitive genomic sequences. The constant methylation status of these sequences has been reported to potentially play a role in the routine upkeep of healthy cells by averting chromosomal instability, translocations, and genetic disruptions. The latter, which may occur through the reactivation of certain transposon-derived sequences that have self-propagation and random site insertion properties, is prevented by hypermethylation [28,33]. Additionally, DNA methylation is one of the most reliable means to transmit epigenetic information across cellular replication [34,35,36]. Thus, maintaining the integrity of DNA methylation patterns is essential for proper cellular function, and disruptions to this process can have significant effects on health and disease. 

Because cytosine is typically paired with guanine, a DNA sequence where several methylated cytosine and guanine pairs come together are known as ‘CpG or CG Islands’, where the highest amount of methylation is present in the genome [37]. CpG islands can be found throughout the genome, and their exact location and frequency can vary depending on the organism and the specific region of the genome [37]. CpG islands frequently occur near the 5′ end of genes (~70%) that contain DNA sequences corresponding to the promoter, untranslated region (5′-UTR), and exon 1 (Figure 2). Unmethylated CpG sites permit the related sequences to be expressed when the required transcriptional activators are available [28,38,39].

The process of DNA methylation is carried out by DNA methyltransferases (DNMT), which transfer a methyl group from S-adenosyl methionine (SAM), a carrier molecule, to the DNA molecule, resulting in the addition of a methyl group to the cytosine base. While several of these enzymes exist, all of them utilize SAM as the carrier molecule. The proteins encoded by the *DNMT3* gene and its variants (*DNMT3A*, *DNMT3B*, regulatory *DNMT3L*) preferentially methylate unmethylated DNA strands and thus carry out a major part of de novo methylation [35]. Meanwhile, the proteins encoded by the *DNMT1* gene methylate DNA whose single strand has already been methylated (hemimethylated DNA) in a preferential fashion [35]. This permits it to maintain the methylation patterns across cellular replication [36,40]. 

To ensure the reliability of DNA methylation, cells have several mechanisms in place to monitor and repair methylation patterns. For example, enzymes of the Ten-Eleven Translocases (TET) family (*TET1*, *TET2*, *TET3*), can remove methyl groups from DNA, and help reverse the de novo methylation process, while other enzymes can recognize and repair damaged or improperly methylated DNA. TET enzymes, which are α-ketoglutarate-dependent dioxygenases, convert 5-methylCytosine (5mC) to 5-hydroxymethylCytosine (5hmC), 5-formylCytosine (5fC), and 5-carboxylCytosine (5caC) in a stepwise fashion [41,42], as part of the normal cytosine methylation cycle (Figure 3). The 5-carboxylcytosine is later removed by the human thymine-DNA glycosylase (hTDG) enzyme, in a process exemplifying “active DNA demethylation” [43,44]. This is immediately followed by the insertion of an unmethylated cytosine residue at the excision site, carried out by the DNA Base Excision Repair (BER) system [45]. The TET-hTDG-BER system is known to ensure that cells can actively and rapidly demethylate specific loci in response to environmental changes, such as cellular stressors. This active demethylation is in contrast to the passive demethylation process which occurs in locations where DNMT1 is not present to methylate DNA during replication [46]. Additionally, 5-hmC, by itself, has been hypothesized to play a role in the regulation of gene expression, given that it is noted to be present in both tissue-specific gene bodies and DNA enhancers, the latter being short regulatory sequences where transcription factors bind. Thus, dysregulation of this tightly controlled active methylation and active demethylation in healthy cells leads to errors that eventually permit the hallmark neoplastic features to manifest [32]. Efforts are underway to generate genome-wide 5-hmC profiles (tissue maps) of cells in various tumors [47].

## 4. DNA Methylation in Low-Grade Gliomas

The utility of studying DNA methylation was first identified in glioblastoma, due to its aggressiveness and poor prognosis. While such studies have begun to include low-grade gliomas (LGGs) as well, literature specific to LGGs remains scarce (Persico et al., 2022), even though there is wide recognition that DNA methylation is likely to play a key role in the next frontier of oncology diagnostics and therapeutics [49,50].

Fundamentally, methylation of a locus typically results in the repression of its expression level, which can then affect the expression level of other genes that are downstream targets. Methylated DNA sequences are less accessible to the cellular machinery that reads the genetic code. For example, if the locus has elements that repress expression (e.g., 5′ regulatory region) of the associated gene(s) (e.g., a DNA damage repair gene), then the methylated locus would become silenced, leading to an increase in gene expression of the associated gene (in this case higher production of DNA damage repair proteins). 

In general, while cancer cells undergo a global loss of DNA methylation (Figure 2), CpG islands of tumor-suppressor genes (TSGs) undergo preferential hypermethylation [28]. The epigenetic silencing of TSGs permits the cancer cell to evade pro-apoptotic changes, proceed with unrestrained cellular replication, display angiogenesis and reduce cellular adhesion, amongst other mechanisms, thus contributing to the classically described hallmarks of cancer cells [28,35,51] (Figure 2). These unique DNA methylation changes are also accompanied by histone modifications, another epigenetic alteration that permits further silencing of TSGs and increased expression of oncogenes [52,53,54], as discussed later in the text. Hypermethylation of tumor suppressor genes is increasingly being explored as a prognostic marker in low-grade gliomas, for instance, testing for MGMT methylation status to predict response to chemotherapy [55]. O6-methylguanine-DNA methyltransferase (*MGMT*) is a protein involved in DNA repair. When the *MGMT* gene locus become methylated (i.e., hypermethylated), the amount of DNA repair across the genome reduces, leading to increased sensitivity to cytotoxic medications, making the tumor more responsive to chemotherapy [33]. Therefore, in gliomas, *MGMT* hypermethylation is associated with a better response to temozolomide, a DNA alkylating agent.

*MGMT* promoter hypermethylation is being increasingly explored as a clinical target in LGGs. It has been recently reported to be a predictor of hypermutation in LGGs at the time of recurrence. Mathur et al. demonstrated in 2020 that methylation-based silencing of *MGMT* expression enhances mutagenic processes caused by temozolomide in LGGs, thus leading to the development of hypermutation in these tumors. Further, analysis of DNA methylome of genes involved in DNA damage repair in the EORTC 22033 trial cohort has demonstrated that patients having a high MGMT-STP27 score, which measures methylation status, prognosticates those patients of IDHmt LGGs who are most likely to benefit from temozolomide chemotherapy [56]. Meanwhile, work from UCSF has demonstrated that temozolomide positively selects for tumor cells with *MGMT* hypermethylation in patients with LGGs lacking DNA mismatch repair (MMR) [57]. Given these and similar findings from the literature, *MGMT* promoter methylation is likely to serve as a useful biomarker for predicting response to therapy and risk of hypermutation at recurrence [56,57,58].

In addition to the involvement of DNA methylation in cellular processes in LGGs, errors in DNA methylation also predispose to mutations. Compared to cytosine (C), methylated cytosine residues (mC) are more prone to deamination, i.e., loss of the amine (-NH2) group, forming thymine residues, which are less likely to be repaired accurately [45]. This mutational event then changes the DNA sequence, which is the primary driver of the sequence of corresponding messenger RNA, leading to abnormalities in structure, quantity, or function in subsequent protein synthesis. Thus, ‘CpG Islands’ are more prone to mutations than human DNA sequences in general. One pertinent example is the glioma CpG island methylator phenotype (G-CIMP), a pattern of genetic changes that includes MGMT methylation, which is often associated with the presence of *IDH1* or *IDH2* gene mutations. G-CIMP, while quite underexplored in LGGs, likely represents a major avenue for future research given that Grade 2 astrocytoma (IDHmt) and oligodendroglioma (IDHmt, 1p/19q codeletion) are both characteristically associated with G-CIMP. This attribute gains importance given that, amongst WHO Grade 2/3 astrocytomas, oligodendrogliomas, and glioblastomas developing from these lower grade entities, IDH1 mutation occurs at codon number 132 in over two-thirds of these, with IDH2 mutations occurring in 6% of them [13]. Given that MGMT resides on chromosome 10, it has been reported that compared to GBM, where at least one copy of chromosome 10 is lost, IDHmt lower-grade gliomas do not lose either copy. Thus, sufficient silencing of the MGMT gene may not occur in these IDHmt gliomas, leading to MGMT expression, followed by remnant capacity for DNA repair. This is the likely cause behind the resistance of IDHmt gliomas to temozolomide chemotherapy, compared to GBM [45]. Additionally, the deletion of 1p36 has been demonstrated to occur in nearly 73% of oligodendrogliomas and 18% of astrocytomas, while the deletion of 19q13.3 chromosome has been found to occur in 73% of oligodendrogliomas and 38% of astrocytomas. 1p/19q-codeletion has been demonstrated to occur in nearly 64% of oligodendrogliomas and 11% of astrocytomas [59,60]. 

Additionally, methylation is known to alter the overall 3-dimensional organization of chromatin protein used for DNA compaction. Chromatin consists of loops or topology-associated domains (TADs), which are normally conserved and maintained across cells [45]. The architecture of TADs has been demonstrated to be disturbed in IDHmt gliomas, causing excessive oncogene and anti-apoptotic factor expression [61,62]. One example is the Cohesin and CCCTC-binding factor (CTCF), whose alteration affects the organization of TADs [45].

DNA methylation has also recently been implicated in the functioning of the Telomerase Reverse Transcriptase (TERT) gene, whose function is visually described in Figure 4. TERT-promoter mutations (TERT-pmt) are known to be amongst the most common and the earliest mutations in the most invasive gliomas [63,64,65,66,67]. TERT mutations have been reported to be closely associated with *IDH1/2* mutations and 1p/19q-codeletion in oligodendroglioma, but less well correlated in astrocytomas [68,69]. It has been hypothesized that TERT promoter mutations enhance the neoplastic potential of tumors with low rates of self-renewal, such as low-grade gliomas [70]. Where methylation additionally plays a role is in the regulation of the TERT gene, whose promoter region has elements called “GC boxes”. These GC-base pair rich DNA sequences preferentially bind to the transcriptional activator SP1, leading to increased gene expression. These GC boxes are closely regulated through DNA methylation [71]. Furthermore, hypermethylation of the TERT promoter region has been demonstrated to be one factor behind the dysregulation of TERT function in cancer cells [72,73,74]. Uniquely, TERT hypermethylated oncological region (THOR), a 433-bp sequence, has been reported to be a region where methylation leads to increased transcriptional TERT activity. It is situated just upstream of the TERT promoter region and contains 52 CpG sites. THOR hypermethylation has been demonstrated to play a role in the pathogenesis and/or outcomes of several pediatric brain tumors, including gliomas [75,76,77].

DNA methylation, within the context of low-grade gliomas, also plays a role in the regulation of the ADP-ribosylation factor-like (ARL) family of genes. The ADP-ribosylation factor (ARF) family of proteins, a part of the RAS superfamily, had been previously demonstrated to play a part in the pathogenesis of both glioblastoma and lower-grade gliomas [78,79,80]. Utilizing the TCGA database, Tan et al. recently identified low expression of ARL9 mRNA, along with ARL9 hypermethylation, which had hitherto been unexplored in LGGs, as positive prognostic factors in LGG [81]. The ARL9 protein expression was reported as correlating with CD8 T-cells in the LGG tissue, indicating the role of ARL9 methylation in tumor immune infiltration [81].

Broad prognostic signatures based on epigenetics have been very recently developed for low-grade gliomas. A two-CpG site DNA methylation signature (GALNT9 and TMTC4, both of whose expressions are highly dependent on methylation) has been recently identified that correlated highly with prognosis, regardless of the age, WHO grade, family history of cancer, and IDH mutation status [82]. Similarly, three methylation-driven genes (ARL9, CMYA5, STEAP3) have been recently identified as independent prognostic factors for survival in LGGs [83].

Overall, DNA methylation is an important mechanism for regulating gene expression in cancer cells, including LGGs, through several pathways (Figure 5). Alterations in DNA methylation lead to changes in gene expression that can result in neoplastic processes. The precise pattern of DNA methylation likely varies between cells of different grades and types of LGGs, being influenced by several factors, most of which are under investigation.

## 5. Overview of Histone Modification

Histones are proteins that DNA is wrapped around to compact DNA in the nucleus. Together, an octamer of histones, with DNA wrapped around it, form a nucleosome, which is the functional unit of chromatin [84].

Histones are traditionally highly conserved across species. Post-translational modification of the histone typically occurs at one end, called the N-terminal tail, and is a significant epigenetic mechanism. This modification could be phosphorylation, ADP ribosylation, methylation, or acetylation, among others [85]. Methylation and acetylation, for example, are processes by which methyl and acetyl groups, respectively, are added to their amino acid residues in an enzyme-dependent fashion. These modifications can also change the expression of a DNA segment, without changing the underlying sequence.

Histone methylation is carried out by enzymes called histone methyltransferases, which transfer a methyl group from S-adenosylmethionine (SAM) to the histone protein. The particular residue that is methylated, and the number of methyl groups added, can vary and can have different effects on gene expression. For example, the addition of a single methyl group to a lysine residue on a histone protein (mono-methylation) can have a relatively mild effect on gene expression, while the addition of three methyl groups to the same residue (tri-methylation) can have a much stronger effect. Typically, methylation causes transcription dysregulation [85]. Figure 6 summarizes the differences in histone modification maps in healthy cells versus neoplastic ones.

Histone acetylation refers to the addition of an acetyl functional group, through a reaction between the hydrogen atom of a hydroxyl (-OH) group and an acetyl (CH3CO) group. This usually occurs on the lysine and arginine residues of histone proteins. Acetylation is carried out by histone acetyltransferases (HATs), while the reverse is carried out by histone deacetylases (HDACs). Acetylation of lysine weakens histone-DNA or inter-nucleosome interactions [86,87], altering chromatin conformation, and facilitating transcription. Conversely, deacetylation diminishes transcription. In normal cells, HATs and HDACs act in a dynamic equilibrium. Dysregulated acetylation, as in cancer cells, usually affects DNA transcription and repair [85].

## 6. Histone Modification in LGGs

Histone modifications have been studied far more in high-grade gliomas, and the advances made there have not yet translated into the field of LGGs, but significant potential for translational research exists here. In particular, in diffuse midline gliomas, the H3K27M alteration has been shown to confer poor prognosis. Here, the H3 subunit, referring to either H3.1 or its variant H3.3, is subject to post-translational modifications, including methylation and acetylation. Typically, in the H3K27M alteration, methionine substitutes lysine at residue 27, resulting in halted post-transcriptional silencing by trimethylation. This modification resembles a gain-of-function mutation that enables the inhibition of polycomb repressive complex 2 (PRC2), as well as an increase in histone hypomethylation [88,89]. Additionally, it has become clear that the H3 variant also matters. H3.1K27M commonly co-occurs with activin-receptor type 1 (ACVR1) and phosphoinositide 3-kinase (PI3K), while the H3.3K27M commonly occurs with deletions of tumor suppressor 53 (TP53) and amplification of platelet-derived growth factor, with the latter shown to be significantly more aggressive and less differentiated [89,90]. Given the shared attributes of precursor cells of origin for LGGs and HGGs, these specific findings need investigation in LGGs as well. 

Central to the advances made in histone modifications in LGGs has been the seminal discovery of IDH mutations as a genetic signature of most LGGs [91]. In IDHmt glioma cells, the disrupted metabolism of 2-hydroxyglutarate is key to their oncogenesis. As opposed to the conversion of isocitrate to alpha-ketoglutarate (α-KG) in IDHwt cells, IDHmt cells convert α-KG to 2-HG at supraphysiologic levels. This results in 2-HG levels several-fold higher than in IDHwt cells [92], with decreased levels of α-KG. 2-HG accumulation is likely a key step in gliomagenesis, which sets the stage for multiple later mutations [91]. 2-HG has been shown to alter DNA repair mechanisms, particularly the homologous recombination (HR) pathway, as well as multiple key cellular metabolic and oxidative pathways [93,94]. With respect to histones, 2-HG accumulation promotes methylation, through the inhibition of Jumonji-C-domain histone demethylases (JHDMs) [91,95,96,97]. These cumulative effects result in the G-CIMP phenotype of LGGs [91]. Further, as in pediatric diffuse gliomas, IDH1 mutations that cause H3K27 or H3K36 methylation have been implicated in progression from LGGs to GBM, i.e., secondary GBM [98].

## 7. Current State of Therapeutics 

Table 2 summarizes ongoing (as of 13 January 2023) clinical trials in IDH-mutant LGGs, which broadly indicate that therapies targeting DNA and histone modification are gaining increasing cognizance. 

Based on the current understanding of the role of epigenetics in LGGs, several potential targets have emerged, albeit with preclinical data. Ongoing and completed trials remain in the early phases, and a long wait for definitive results is anticipated.

### 7.1. Therapeutics Targeting IDH1/2 Mutations

Given the central role of the IDH mutation in LGGs as a driver mutation and its role in downstream epigenetic modification, it is worth discussing attempts at targeting IDH1/2 mutations in LGGs. Data from completed clinical trials targeting IDH, all of which have been phase I trials, are summarized in Table 3.

IDH mutations, as well as the downstream accumulation of 2-HG [91], have been the focus of some of the earliest attempts for translating epigenetics from bench-to-bedside in LGGs, although preclinical results have been mixed. While Rohle et al. found reduced 2-HG levels and slowed growth in glioma xenografts by AG-5198 in 2013 [105], in later years, subsequent groups failed to show encouraging outcomes, be it regarding tumor size, DNA, or histone methylation [91]. In mouse IDHmt models, AG-120, a successor of AGI-5198, was found to be highly effective, leading to demonstrably lower levels of 2-HG, and reduced cell proliferation [106]. Later investigated drugs of the same class include BT142 and GB10, with only BT142 showing tumor growth inhibition in xenografts [107]. 

With discoveries that 2-HG greatly contributes to glioma immune escape and immunosuppressive mechanisms, immunotherapy targeting IDH mutations has been another promising avenue [91,108]. IDHmt vaccines targeting specific epitopes demonstrated efficacy in a glioma model [109]. More recently, Kadiyala et al. demonstrated significantly improved outcomes in IDH1-mt gliomas in mice, with the administration of a targeted inhibitor, either alone, or with radiation and TMZ [91,110].

Similarly, the effect of 2-HG on the HR pathway of DNA repair has been investigated [91,94]. IDHmt LGG cells have defective DNA repair, especially in the HR pathway, which is the most preferred mechanism of repair in most cells [93]. This, along with its backup mechanism, the alternative end-joining pathway of DNA repair, is highly dependent on poly(ADP-ribose) polymerase (PARP) [94]. Thus, PARP inhibitors are under investigation, particularly in combination with radiotherapy (RT) or temozolomide (TMZ). Wang et al. and Higuchi et al., in their preclinical models, demonstrated that PARP inhibition’s efficacy may be enhanced by combination with TMZ or RT [111,112]. Recent clinical trials include a phase II trial investigating PARP inhibitors (Olaparib) alone for IDHmt advanced gliomas (NCT03212274), and a phase II trial investigating Olaparib in recurrent IDHmt gliomas (NCT03561870). Combinations of PARP inhibitors are also being investigated—NCT03749187 is a trial of BGB-290, a novel PARP inhibitor in combination with TMZ for IDHmt gliomas of all grades, while NCT03914742 is investigating the same combination for recurrent IDHmt gliomas, and NCT03991832 is investigating Olaparib in combination with a checkpoint inhibitor, Durvalumab [91] (Table 2).

Further, some hypothesized therapeutic pathways involve exploiting metabolic and apoptotic vulnerabilities in IDHmt cells [91,94]. However, the caveat remains that some of these results are from IDHmt GBM isolates, or isolates of other tumors, not from LGG-specific cell cultures. Tateishi et al. demonstrated that IDHmt glioma cells had lowered NAD+ levels, a crucial cofactor for cellular metabolism. Further, their team found that these cells were sensitive to inhibitors of nicotinamide phosphoribosyl transferase (NAMPT), an enzyme necessary for NAD+ synthesis [113]. NCT02702492 is an ongoing Phase I trial that is investigating KPT-9274, one such agent, in IDHmt solid tumors. In IDHmt tumor models, the presence of raised 2-HG levels was shown to trigger apoptosis by suppressing BCL-2, causing altered mitochondrial metabolism and apoptosis [65,91,94]. Another group of authors found that ABT263, a BCL-2 and BCL-xL inhibitor, was lethal to IDHmt glioma cells [94,114]. One avenue includes altering the production of 2-HG, by halting its production from α-KG. α-KG is produced from glutamate, and reducing glutaminase activity has been shown to reduce the growth and increase the sensitivity of IDHmt glioma cells to radiation [91]. Finally, IDHmt glioma cells have been shown to specifically exhibit greater levels of Notch ligand delta-like 3 (DLL3) RNA and were sensitive to anti-DLL3 antibodies [115]. The caveat to these advances, besides the fact that they are at the preclinical level, remains that most results are from studies on GBM-derived cells, or even IDHmt cells from other cancers. Regardless, they may provide some cause for cautious optimism.

### 7.2. Therapeutics in DNA Methylation, Histone Modification, and Other Domains of Epigenetics in LGGs 

DNA demethylating agents, or DNA methyltransferase inhibitors (DNMTIs), were investigated early on [91], given the hypermethylated phenotype of IDHmt gliomas. Preclinical glioma models investigating long-term 5-azacitidine and decitabine demonstrated significant tumor growth inhibition [116,117], which another group of authors demonstrated to be enhanced by combination with temozolomide [118]. However, these results have not yet been translated to the clinical setting. In a clinical trial of 12 patients with IDHmt recurrent gliomas (astrocytic or oligodendroglial histology), 5-azacitidine demonstrated minimal activity [119]. Current ongoing trials include those testing 5-azacytidine, either as a single agent or in combination with IDHmt inhibitors (NCT03666559, NCT03684811), while another phase I trial is ongoing to evaluate ASTX727, a combination of decitabine and a cytidine deaminase inhibitor in recurrent or progressive IDHmt gliomas (NCT03922555) [91] (Table 2).

Despite prior knowledge of their presence, the role of histone modifications in LGG therapeutics has come to the fore only in recent years [45]. The clinical utility of histone modification in LGGs is best exemplified through the Histone Deacetylase (HDAC) inhibitors. Panobinostat achieved feasibility in Phase I trials using glioma cells, and FDA approval for off-label use for diffuse gliomas [89]. Its combinations with the proteasome inhibitor marizomib have also been explored in preclinical studies [89,120] (Cooney et al., 2020; Kilburn et al., 2018). Finally, it has also been demonstrated that valproate, the well-known antiepileptic, and Panabinostat both inhibit IDHmt glioma cell lines [91].

Finally, Bromodomain and Extra-Terminal Motif (BET) inhibitors are a target of promise. BET proteins are key in epigenetic regulation, and promote the expression of multiple oncogenes [91]. IDHmt glioma cells have been found to be sensitive to two BET inhibitors (JQ1 and GS-626510) [121].

## 8. Conclusions

Several prognostic biomarkers and potential therapeutic targets may be identified in cellular structures and processes associated with DNA methylation and histone modification in low-grade gliomas. Diagnostic and/or therapeutic targeting of MGMT promoter methylation, TET-hTDG-BER pathway, G-CIMP association, PARP inhibition, IDH and 2-HG-associated processes, TERT mutation and ARL9-associated pathways, DNA Methyltransferase (DNMT) inhibition, Histone Deacetylase (HDAC) inhibition, BET inhibition, and CpG site DNA methylation signature, along with others, present exciting avenues for translational research. However, much of the evidence remains restricted to preclinical studies, warranting further investigation to demonstrate true clinical utility.

## Figures and Tables

**Figure 1 cancers-15-01342-f001:**
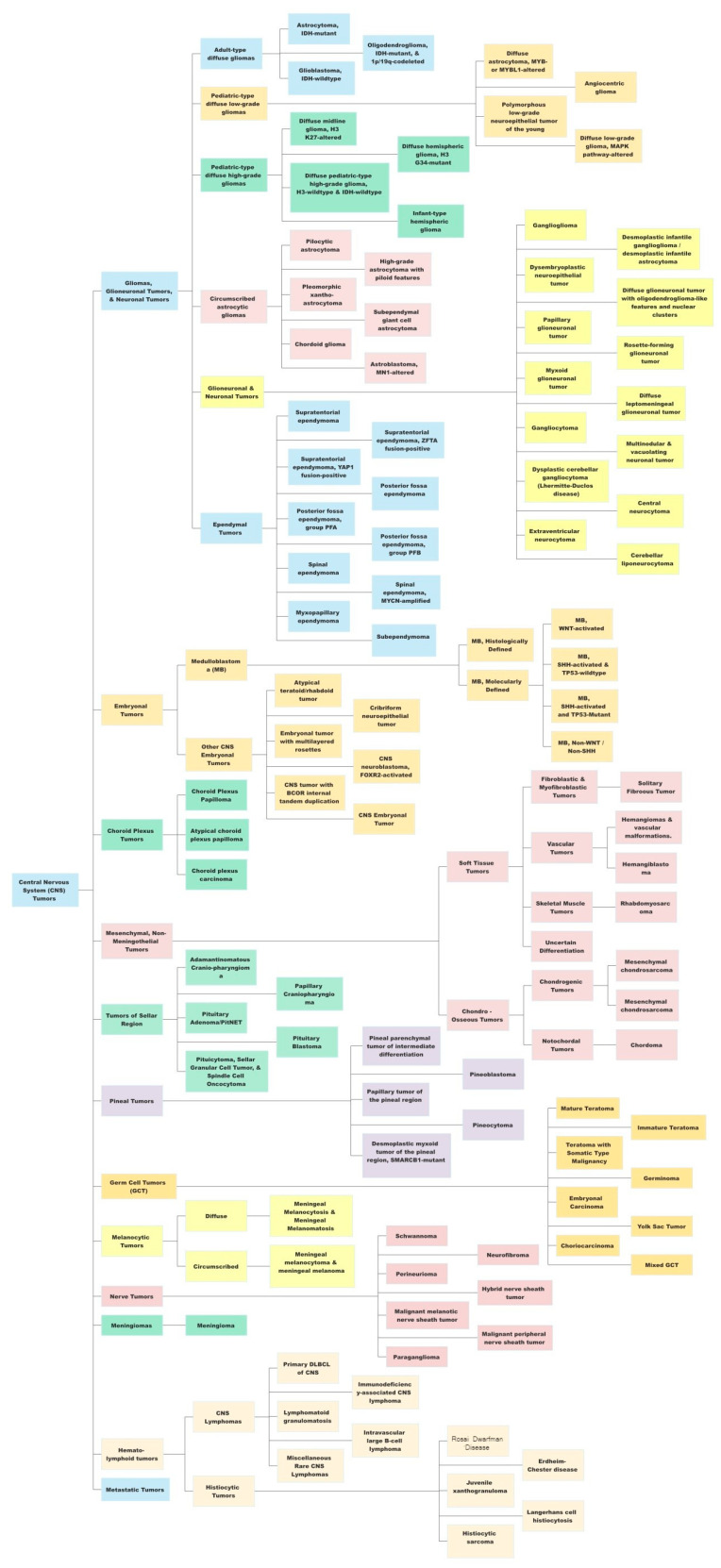
A summary view of the World Health Organization (WHO) 2021 classification of central nervous system (CNS) tumors. This original figure has been created using data available from the WHO CNS5 publication.

**Figure 2 cancers-15-01342-f002:**
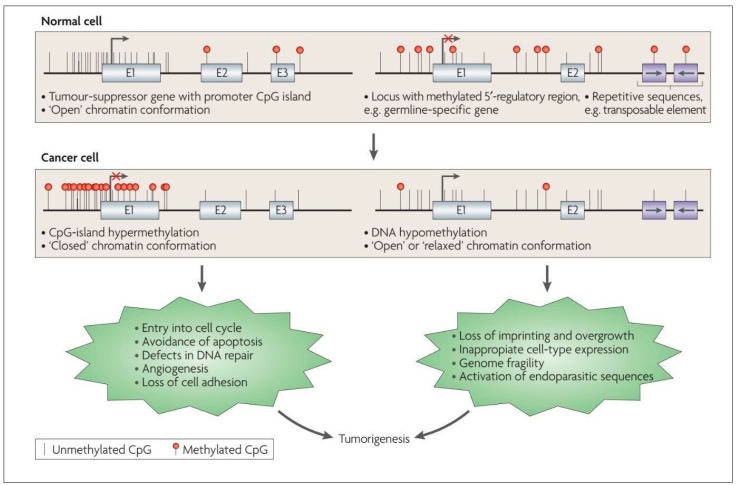
Altered DNA methylation and its downstream impact in the cancer cell. Reproduced with permission from [28].

**Figure 3 cancers-15-01342-f003:**
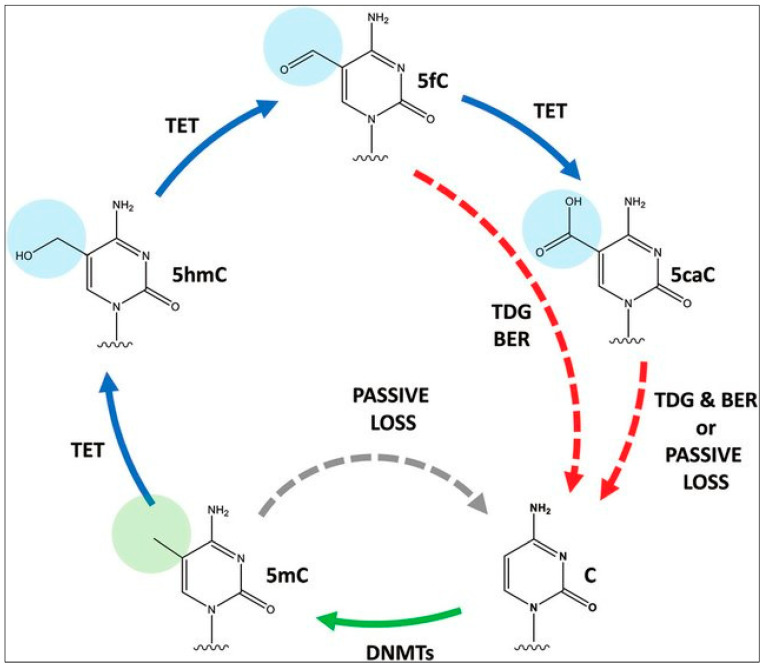
Cytosine methylation and demethylation cycle. C, cytosine; 5mc, 5-methylCytosine; 5hmC, 5-hydroxymethylCytosine; 5fc,5-formylCytosine; 5caC, 5-carboxylCytosine, TDG, thymine-DNA glycosylase; BER, Base Excision Repair, TET, Ten-Eleven Translocases, DNMT, DNA Methyltransferases. Reproduced with permission from [48].

**Figure 4 cancers-15-01342-f004:**
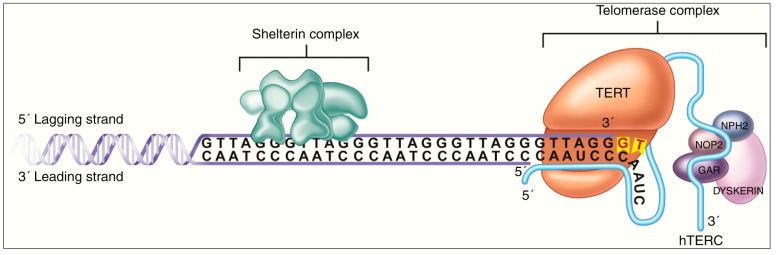
Mechanism of action of TERT enzyme, whose regulation is impacted by methylation of promoter and upstream THOR sequence. In the figure, TERT accesses the telomere complex at the terminal end of the DNA strand, through the Shelterin complex. It then catalyzes the addition of telomere repeat segments with the help of the hTERC enzyme, in a structure called Telomerase Complex. The latter’s function of telomere elongation works against the routine telomere shortening that occurs during DNA replication. Figure reproduced under Creative Commons Attribution-Noncommercial 4.0 License from [74].

**Figure 5 cancers-15-01342-f005:**
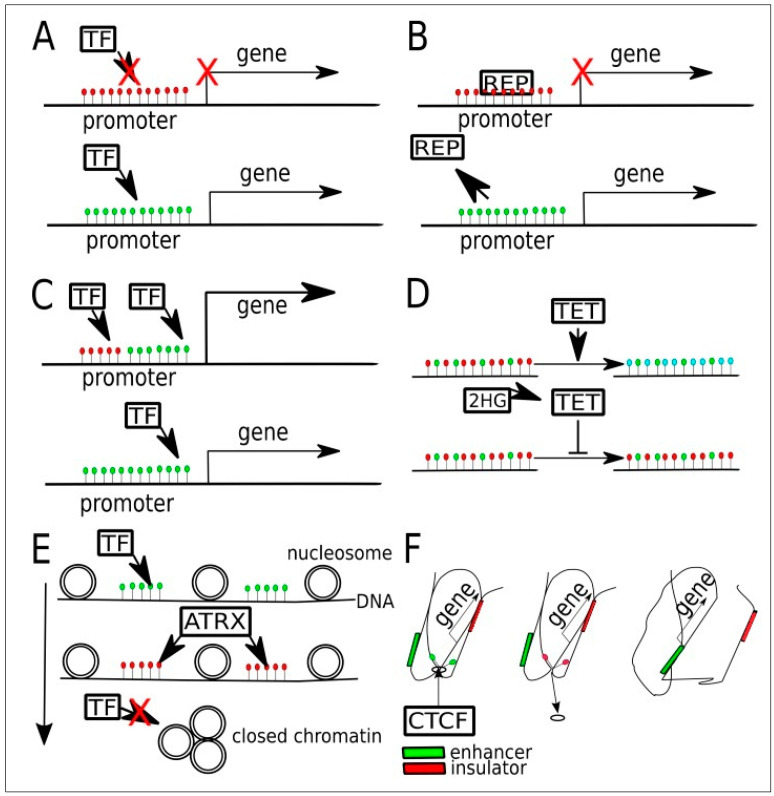
Potential targets in the various pathways where DNA methylation plays a role in regulating gene expression in gliomas. (Green dots are unmethylated Cytosine, red dots are 5-methylCytosine; blue dots are 5-hydroxymethylcytosine). (**A**) Promoter hypermethylation may prevent the binding of transcriptional factors (TF), i.e., activator, leading to gene silencing. (**B**) In some other cases, a hypermethylated promoter may bind to the transcriptional repressor (REP) preferentially. When active demethylation occurs, REP is unable to bind and gene expression occurs. (**C**) In another gene, there may occur binding by two transcriptional factors (TFs), one to a methylated sequence and another to an unmethylated sequence. (**D**) In normal cells, TET enzymes convert 5mc to 5hmc and later into 5cac for maintenance purposes. When 2-Hydroxyglutarate (2-HG), a byproduct of mutant IDH enzymes, inhibits TET, a state of global hypermethylation occurs. (**E**) Relationship between DNA methylation and chromatin compaction. The latter is regulated by chromatin chaperones that are in turn affected by DNA methylation, histone methylation, and histone acetylation. ATRX binding to methylated gene sequences leads to an increased proportion of heterochromatin, thus reducing the binding of transcriptional factors (TFs) to DNA. (**F**) When CTCTF binding sites on the genome are methylated, then CTCF is unable to bind, leading to alteration in chromatin compaction. This causes an exchange of an insulator by an enhancer near the said sequence. Figure reproduced, with color correction, under Creative Commons Attribution-Noncommercial 4.0 license from [45].

**Figure 6 cancers-15-01342-f006:**
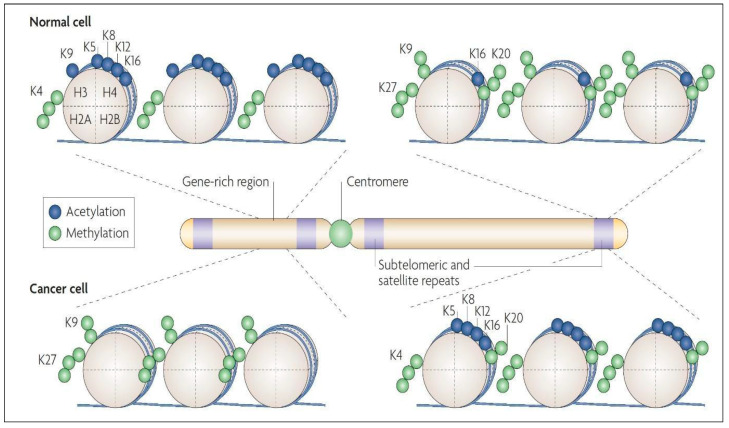
Histone modification maps for a typical chromosome in normal and cancer cells. In normal cells, DNA sequences that include the promoters of tumor-suppressor genes have more histone modification marks associated with active transcription, such as acetylation of H3 and H4 lysine residues (e.g., K5, K8, K9, K12, and K16) along with trimethylation of K4 residue of H3 protein. The normal cell also has DNA repeats and other heterochromatic regions having repressive histone marks, such as trimethylation of K27 residue and dimethylation of the K9 residue of H3, and trimethylation of K20 of H4. In cancer cells, there is a loss of the “active” histone marks on promoters of tumor-suppressor genes, leading to a tighter chromatin configuration. Additionally, the neoplastic cell has a loss of repressive marks at subtelomeric DNA and other repeat regions, causing a more “relaxed” chromatin conformation in these regions. Figure reproduced with permission from [28].

**Table 1 cancers-15-01342-t001:** Status of gliomas in the fifth edition of the WHO Classification of Tumors of the Central Nervous System (WHO CNS5). Adapted under Creative Commons Attribution-Noncommercial-Share Alike 4.0 License from [23]. Available from: https://www.ijpmonline.org/text.asp?2022/65/5/5/345057. Accessed on 15 December 2022.

Gliomas, Glioneuronal and Neuronal Tumors	WHO Grade	Remarks
**Ependymal Tumors**		
**Adult-type diffuse gliomas**		
Astrocytoma, IDH-mutant	2, 3, 4	“Diffuse” and “anaplastic” are terms no longer used; no tumor exists now that is called ”astrocytoma, IDH-wild type”.
Oligodendroglioma, IDH-mutant and 1p/19q-codeleted	2, 3	Similar grading approaches to WHO CNS4 (2016); tumor type ”oligoastrocytoma” deleted.
Glioblastoma, IDH-wildtype	4	Terms such as ”multiforme” and ”Glioblastoma, IDH mutant” were removed from WHO CNS5. Three subtypes, namely giant cell type, gliosarcoma, and epithelioid type, are still discussed in the WHO CNS5 text but removed from the classification.
**Pediatric-type diffuse low-grade gliomas (pDLGG)**		
Diffuse astrocytoma, MYB- or MYBL1 altered	1	Newly recognized tumor type.
Angiocentric Glioma	1	First added in WHO 2007 classification under “neuroepithelial tumors”, later moved in WHO 2016 classification to “other gliomas” and in WHO 2021 moved to ”pDLGG”
Polymorphous low-grade neuroepithelial tumor of the young	1	Newly recognized tumor type.
Diffuse low-grade glioma, MAPK altered	Unassigned	Newly recognized tumor type.
**Pediatric-type diffuse high-grade gliomas (HGG)**		
Diffuse midline glioma (DMG), H3 K27-altered	4	Revised nomenclature: H3K27-altered instead of H3K27-mutant to recognize additional mechanisms.
Diffuse hemispheric glioma, H3 G34-mutant	4	Newly recognized tumor type.
Diffuse pediatric-type HGG, H3-wildtype and IDH-wildtype	4	Newly recognized tumor type.
Infant-type hemispheric glioma	Unassigned	Newly recognized tumor type.
**Circumscribed astrocytic gliomas**		
Pilocytic astrocytoma	1	-
High-grade astrocytoma with piloid features	Unassigned	Newly recognized tumor type.
Pleomorphic xanthoastrocytoma	2, 3	The term ”anaplastic” is eliminated.
Subependymal giant cell astrocytoma	1	-
Chordoid glioma	2	Revised nomenclature – location modifier of ”third ventricle” dropped.
Astroblastoma, MN1 altered	Unassigned	Revised nomenclature – genetic modifier added for specificity (MN1 altered).
**Glioneuronal and neuronal tumors**		
Ganglioglioma	1	-
Desmoplastic infantile ganglioglioma/astrocytoma	1	-
Dysembryoplastic neuroepithelial tumor	1	-
Diffuse glioneuronal tumor with oligodendroglioma-like features and nuclear clusters	Unassigned	Newly recognized tumor type.
Papillary glioneuronal tumor	1	-
Rosette-forming glioneuronal tumor	1	-
Myxoid glioneuronal tumor	1	Upgraded from a provisional status in 2016 to a distinct tumor type.
Diffuse leptomeningeal glioneuronal tumor (DLGNT)	2, 3	Three subtypes added: DLGNT-1q-gain, DLGNT-MC-1, and DLGNT-MC-2.
Gangliocytoma	1	-
Multinodular and vacuolating neuronal tumor	1	New tumor type in WHO 2021, after being upgraded from a mere pattern of ganglion cell tumors in WHO 2016.
Dysplastic cerebellar gangliocytoma (Lhermitte-Duclos disease)	1	-
Central neurocytoma	2	-
Cerebellar liponeurocytoma	2	-
Extraventricular neurocytoma	2	-

**Table 2 cancers-15-01342-t002:** Ongoing clinical trials in IDH-mutant LGGs. Adapted under Creative Commons Attribution 4.0 International (CC BY 4.) License from [91].

NCT Number	Phase	Population	Study Medication	Current Status *
NCT04164901	3	Residual or recurrent IDH1/2-mt grade 2 gliomas	Vorasidenib (AG-881) versus placebo	Active, not recruiting
NCT03684811	1/2	Advanced IDH1-mt gliomas, GBM, other solid tumors (hepatocellular carcinoma; bile duct carcinoma; cholangiocarcinoma; other hepatobiliary carcinomas; chondrosarcoma)	FT-2102 with azacitidine (for gliomas)	Completed
NCT03991832	2	Advanced IDHmt gliomas, other solid tumors (cholangiocarcinoma and others)	Durvalumab and Olaparib	Recruiting
NCT03557359	2	Recurrent/progressive IDH-mut gliomas	Nivolumab	Active, not recruiting
NCT03718767	2	IDHmt gliomas	Nivolumab	Recruiting
NCT03212274	2	IDH1/2-mt gliomas (WHO grade 2, 3, GBM, recurrent), other solid tumors (cholangiocarcinoma, others)	Olaparib	Recruiting
NCT03561870	2	Recurrent IDHmt gliomas, high-grade gliomas	Olaparib	Completed
NCT03749187	1	IDH1/2-mt gliomas	PARP inhibitor (BGB-290) and TMZ	Recruiting
NCT03914742	1/2	Recurrent IDH1/2-mt gliomas	PARP inhibitor (BGB-290) and TMZ	Active, not recruiting
NCT03666559	2	Recurrent IDH1/2-mt gliomas	Azacitidine	Recruiting
NCT03922555	1	Recurrent/progressive non-enhancing IDHmt gliomas	ASTX727 (cedazuridine + cytidine antimetabolite decitabine)	Recruiting

NCT—National Clinical Trials, IDH—isocitrate dehydrogenase; IHDmt—IDH mutant, GBM—glioblastoma * As of 13 January 2023.

**Table 3 cancers-15-01342-t003:** Completed clinical trials with IDH-targeted therapies in glioma cells. Adapted under Creative Commons Attribution 4.0 International (CC BY 4.0) License from [91].

Study	Drug	Population	Key Findings	Adverse Events (>10% Patients)
Mellinghoff et al., 2020 [99]	Ivosidenib (AG-120)	Advanced IDH1-mt solid tumors 35 non-enhancing recurrent gliomas, 31 enhancing recurrent gliomas	500mg once daily selected for expansion cohort DCR 88% vs. 45%; median PFS 13.6 vs. 1.4 months in non-enhancing vs. enhancing cohort	No DLT Headache, fatigue, nausea, vomiting, seizure, diarrhea, aphasia, hyperglycemia, neutropenia, depression, hypophosphatemia, paresthesia
Mellinghoff et al., 2021 [100]	Vorasidenib (AG-188)	Advanced IDH1/2mt solid tumors 22 non-enhancing recurrent gliomas, 30 enhancing recurrent gliomas	Recommended dose < 100 mg in gliomas Non-enhancing glioma: ORR 18% (1 PR; 3 minor responses; 17 SD) Enhancing glioma: ORR 0% (17 SD) Median PFS: 36.8 vs. 3.6 months in non-enhancing vs. enhancing groups	DLT (grade 2 ALT/AST increase) in 5 pts at 100 mg dose levels Headache, AST/ALT increase, fatigue, nausea, seizure, hyperglycemia, vomiting, constipation, dizziness, neutropenia, cough, diarrhea, aphasia, hypoglycemia
Mellinghoff et al., 2019 [101]	Perioperative Ivosidenib (*n* = 13) or Vorasidenib (*n* = 14)	Recurrent non-enhancing IDH1-mt LGGs undergoing craniotomy	2-HG concentration 92% (ivosidenib), 92.5% (vorasidenib) lower in resected tumor tissue of treated patients	Diarrhea, constipation, hypocalcemia, nausea, anemia, hyperglycemia, pruritus, headache, fatigue
Wick et al., 2021 [102]	BAY-1436032	Advanced IDH1-mt solid tumors 26 LGG astrocytoma, 13 LGG oligodendroglioma, 16 GBM	1500 mg twice daily selected for expansion cohorts LGG: ORR 11% (1 CR; 3 PR; 15 SD) GBM: ORR 0%, SD 29%. PFS-rate at three months: 0.31 vs. 0.22 in LGG vs. GBM	No DLT Fatigue, dysguesia
Natsume et al., 2019 [103]	DS-100b	Recurrent/progressive IDH1-mt glioma	125–1400 mg twice daily Non-enhancing glioma (*n* = 9): 2 minor responses; 7 SD Enhancing glioma (*n* = 29): 1 CR; 3 PR; 10 SD	DLT (grade 3 WBC decrease) at 1000mg twice daily Skin hyperpigmentation, diarrhea, pruritus, nausea, rash, headache
Platten et al., 2021 [104]	IDH1-vac	Newly diagnosed IDHmt grade 3/4 astrocytomas	93.3% IDH1-vac induced immune response 3 years PFS: 63%, 3 years OS: 84%	No RLTs Mild site reactions

2-HG—2-Hydroxyglutarate; ALT—alanine transaminase; AST—aspartate transaminase; CR—complete response; DCR—disease control rate; DLT—dose-limiting toxicity; GBM—glioblastoma; IDH—isocitrate dehydrogenase; LGG—low-grade glioma; ORR—objective response rate; OS—overall survival; PFS—progression-free survival; PR—partial response; RLT—regime-limiting toxicity; SD—stable disease; WBC—white blood cells.

## Data Availability

Not Applicable as no datasets generated in the work.

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
