# Peer review of "DNA Methylation and Histone Modification in Low-Grade Gliomas: Current Understanding and Potential Clinical Targets"

_cancers, 2023, doi:10.3390/cancers15041342_

Round 1

Reviewer 1 Report

This is an exhaustive review on the issue of the role of DNA methylation and histone modification in low grade gliomas, which is still unknown, at least with respect to the clinical implications.

Fig and tables are clear and the reference list is adequate.

I suggest some minor changes to improve the manuscript:

- delete the phrase at lines 56-57, as the mention of oligoastrocytomas, that were already eliminated in the WHO 2016, is misleading

-In line 113 add "ependymomas" to medulloblastomas

-In line 145 add "in particular astrocytomas"

Reviewer 2 Report

This review would like to be a summary of the knowledge about DNA methylation and histone alterations on low-grade gliomas, particularly focused on the molecular characterization of these tumors following the new WHO classification (CNS5), in order to recognize prognostic factors and potential news target to apply on the clinical management and treatment of the patients.  

The manuscript contains valuable and new data which are of interest for the readers, cover many aspects of gliomagenesis, suggesting new therapeutic approaches, but there are many inaccuracies, some mistake and the paper results difficult to read, confusing and unclear. So, in my opinion, this review has to be improved to be suitable for the neurooncologists on their activity with glioma patients.

Some observations:

·         Already in the “Introduction” the Authors maintain a mix of the previous and last WHO classification.

·         On the chapter 2 (Current Status of LGGs in WHO Classification) there are old data and some flaw as well about codeletion 1p-19q, or about histological parameters for grading, citing a reference that is not reported, and the part on molecular makers is rather confusing.

·         The chapter 3-4-5: should be updated and could be of interest to develop a part on the new data on DNA methylation profile and their meaning on the classifying CNS tumors.

·         The chapter 7 is not really about histone modification in LGG.

·         On the chapter 8, the Diffuse midline glioma, H3 K27-altered, is a high-grade glioma (WHO grade 4), these histone alterations are not only a prognostic factor.

·         Finally, on the references, number 21/22 and 23/24 are double.

Author Response

Please see the attachment (page 3 and 4).
